# Female Reproductive Factors and the Risk of Bronchiectasis: A Nationwide Population-Based Longitudinal Study

**DOI:** 10.3390/biomedicines10020303

**Published:** 2022-01-28

**Authors:** Bumhee Yang, Dong-Hwa Lee, Kyungdo Han, Hayoung Choi, Hyung Koo Kang, Dong Wook Shin, Hyun Lee

**Affiliations:** 1Division of Pulmonary and Critical Care Medicine, Department of Internal Medicine, Chungbuk National University Hospital, Chungbuk National University College of Medicine, Cheongju 28644, Korea; ybhworld0415@gmail.com; 2Division of Endocrinology and Metabolism, Department of Internal Medicine, Chungbuk National University Hospital, Chungbuk National University College of Medicine, Cheongju 28644, Korea; roroko@hanmail.net; 3Department of Statistics and Actuarial Science, Soongsil University, Seoul 06978, Korea; hkd917@naver.com; 4Division of Pulmonary, Allergy, and Critical Care Medicine, Department of Internal Medicine, Hallym University Kangnam Sacred Heart Hospital, Hallym University College of Medicine, Seoul 07441, Korea; hychoimd@gmail.com; 5Division of Pulmonary and Critical Care Medicine, Department of Internal Medicine, Inje University Ilsan Paik Hospital, Inje University College of Medicine, Goyang 10380, Korea; inspirit26@gmail.com; 6Department of Family Medicine & Supportive Care Center, Samsung Medical Center, Sungkyunkwan University School of Medicine, Seoul 06351, Korea; dwshin.md@gmail.com; 7Department of Clinical Research Design and Evaluation/Digital Health, Samsung Advanced Institute of Health Science and Technology (SAIHST), Sungkyunkwan University, Seoul 06351, Korea; 8Division of Pulmonary Medicine and Allergy, Department of Internal Medicine, Hanyang University College of Medicine, Seoul 04763, Korea

**Keywords:** female, sex hormone, reproductive period, menarche, menopause, bronchiectasis

## Abstract

Although the oestrogen level is thought to be involved in the occurrence of bronchiectasis, limited data are available on the relationship between female reproductive factors and the risk of bronchiectasis. We performed a population-based retrospective cohort study of 959,523 premenopausal women and 1,362,401 postmenopausal women without a previous history of bronchiectasis who participated in a health screening exam in 2009 in South Korea. In premenopausal women, compared with a later age at menarche (≥16 years), an earlier menarche (<12 years) was associated with a reduced risk of bronchiectasis with an adjusted hazard ratio (aHR) (95% confidence interval (CI)) of 0.74 (0.67–0.81). However, there were no significant associations between other reproductive factors (breastfeeding, parity, or oral contraceptive use) and the risk of bronchiectasis. In postmenopausal women, the risk of bronchiectasis (aHR (95% CI)) was lower in those with an earlier menarche (0.79 (0.72–0.87) for <12 years vs. ≥16 years), a later menopause (0.90 (0.84–0.96) ≥55 years vs. <40 years), and a longer reproductive period (0.90 (0.86–0.94) for ≥40 years vs. <30 years). There was no significant relationship between parity and the risk of bronchiectasis. Although breastfeeding <1 year (aHR (95% CI) = 0.92 (0.87–0.97) for <0.5 years and 0.93 (0.88–0.97) for 0.5–1 years) and oral contraceptive use <1 year (0.97 (0.94–0.99)) reduced the risk of bronchiectasis, hormone replacement therapy ≥5 years increased the risk of bronchiectasis (1.24 (1.18–1.30)). Female reproductive factors are risk factors for developing bronchiectasis, showing a higher risk associated with shorter endogenous oestrogen exposure regardless of the menopausal status.

## 1. Introduction

In adults, women generally are more vulnerable than men to the development of airway diseases [1,2,3,4] and have a higher susceptibility to the damaging effects of noxious particles (smoking, air pollution, etc.) in the respiratory system [3,5,6]. Female sex hormones including 17ß-estradiol have been suggested to contribute to this sex disparity and many previous studies have evaluated the associations between female sex hormones and the development of chronic airway diseases [1,2,7,8,9,10,11,12,13,14].

Despite several conflicting results, those previous study findings have provided novel insights into the link between female sex hormones and many chronic respiratory diseases [1,8,9,10,11,12,13,14]. In non-cystic fibrosis bronchiectasis (hereafter bronchiectasis), a few previous studies found a sex disparity in prevalence; the prevalence of bronchiectasis in women was affected by their reproductive span [2,4,15,16,17,18,19,20,21,22,23]. Although those results suggested a close relationship between the lifetime oestrogen exposure and the development of bronchiectasis, no study has considered comprehensive female reproductive factors (age at menarche, age at menopause, parity, breastfeeding, oral contraceptive (OC) use, and hormone replacement therapy (HRT)) [2,4,15,16,17,18,19,20,21].

In this study, we evaluated the connections between female reproductive factors and the development of bronchiectasis. In our analyses, we used a population-based longitudinal cohort and several female reproductive factors.

## 2. Materials and Methods

### 2.1. Study Population and Design

We gathered our data from the Korean National Health Insurance Service (NHIS) database [24]. Korea has a single-payer universal health system; the NHIS maintains the claims data on all reimbursed inpatient and outpatient visits, procedures, and prescriptions including data from annual (all Koreans 40 years or older and all employees regardless of age) or biennial (workers in jobs requiring physical labour) health screening exams provided free of charge by the Ministry of Health and Welfare [25,26]. The health screening exams include the National Cancer Screening Program, which screens all Korean women aged 40 or older biennially for breast cancer [27]. During that screening program, all participants are asked to complete a self-administered questionnaire on reproductive factors (age at menarche, age at menopause, parity, breastfeeding, OC use, and HRT).

This study initially included 3,109,506 females older than 40 years who received a health screening exam in 2009 (index year). After excluding 314,529 participants with an uncertain menopause status, our study population contained 1,069,475 premenopausal and 1,725,502 postmenopausal participants. Among the 1,069,475 premenopausal participants, we identified 959,523 eligible participants after excluding those: (1) missing information for at least one variable (*n* = 99,644); (2) diagnosed with cystic fibrosis or congenital bronchiectasis (*n* = 85); (3) diagnosed with bronchiectasis before the enrolment period (*n* = 8044); (4) diagnosed with bronchiectasis within one year after enrolment (*n* = 1734); or (5) who died within one year of enrolment (*n* = 445). Among the 1,725,502 postmenopausal participants, we identified 1,362,401 eligible participants after excluding those: (1) missing information for at least one variable (*n* = 320,845); (2) diagnosed with cystic fibrosis or congenital bronchiectasis (*n* = 115); (3) diagnosed with bronchiectasis before the enrolment period (*n* = 31,556); (4) diagnosed with bronchiectasis within one year after enrolment (*n* = 6689); or (5) who died within one year of enrolment (*n* = 3896) (Figure 1). The participants were tracked from the index year until the date of a bronchiectasis diagnosis, death, or 31 December 2018.

Our study protocol was approved by the Institutional Review Board of Chungbuk National University Hospital (No. 2021-08-016). The requirement for informed consent was waived because the NHIS database uses an anonymous patient identification system.

### 2.2. Exposure

The main exposures were age at menarche in the premenopausal women and reproductive duration in the postmenopausal women. The data on the age at menarche and the age at menopause were obtained from a self-administered questionnaire. The age at menarche was categorised as <12 years, 12–14 years (≥12, <14), 14–16 years (≥14, <16), and ≥16 years. The age at menopause was categorised as <40 years, 40–45 years (≥40, <45), 45–50 years (≥45, <50), 50–55 years (≥50, <55) and ≥55 years. The reproductive period was calculated as the interval between the age at menarche and the age at menopause [28]. The reproductive period was categorised as <30 years, 30–35 years (≥30, <35), 35–40 years (≥35, <40), and ≥40 years.

Other exposures included parity, breastfeeding, and OC use in all women and HRT in postmenopausal women. The data on parity, lifetime breastfeeding history, OC use, and HRT were also obtained from a self-administered questionnaire. The parity was categorised as none, one child, or multiparous (≥2 children). The lifetime breastfeeding history was categorised as none, <0.5 years, 0.5–1 (≥0.5, <1) years, and ≥1 year. The duration of HRT was categorised as none, <2 years, 2–5 (≥2, <5) years, and ≥5 years. The duration of OC use was categorised as none, <1 year, and ≥1 year [28].

### 2.3. Outcome

The main study outcome was incidence of bronchiectasis. Bronchiectasis was defined as claims under the International Statistical Classification of Diseases and Related Health Problems, 10th revision (ICD-10) diagnosis code J47 (bronchiectasis) without a concomitant diagnosis of cystic fibrosis (E84), as used in our previous studies [29,30,31,32,33].

### 2.4. Covariates

The body mass index (BMI) was calculated as the weight of a participant in kilograms divided by the square of height in meters. The smoking status was classified as non-smoker, ex-smoker, and current smoker. Regular exercise was defined as moderate physical activity for more than 30 min on more than 5 days of the past week [34]. Comorbidities (hypertension, diabetes mellitus, dyslipidaemia, strokes, cardiovascular disease, asthma or chronic obstructive pulmonary disease (COPD), and tuberculosis) were collected using ICD-10 codes in the medical records of the participants [29,30,31,32,33,34,35]. The household income was categorised into quartiles based on the insurance premium levels (in Korea, insurance premiums are determined by the income level) with those covered by medical aid (poorest 3%) being merged into the lowest income quartile [28,34].

### 2.5. Statistical Analysis

The continuous variables were presented as mean  ±  standard deviation (SD) and the categorical variables were presented as a number and percentage. The incidence rate of bronchiectasis was expressed as the number of events per 1000 person/years. Cox proportional hazard regression analyses were conducted to evaluate the associations between the various reproductive factors and the incidence of bronchiectasis. The baseline demographics, metabolic comorbidities that might interact with the female reproductive factors (diabetes mellitus and dyslipidaemia), and pulmonary comorbidities that might affect the development of bronchiectasis (e.g., asthma [36,37], COPD [38], and tuberculosis [39]) were considered in the multivariable analyses. For premenopausal women, the multivariable model was adjusted for age, BMI, smoking history, regular exercise, income, age at menarche, parity, duration of breastfeeding, duration of OC use, and comorbidities (diabetes mellitus, dyslipidaemia, asthma or COPD, and tuberculosis). For postmenopausal women, the multivariable model was adjusted for age, BMI, smoking history, regular exercise, income, reproductive period, parity, duration of breastfeeding, duration of OC use, duration of HRT, and comorbidities (diabetes mellitus, dyslipidaemia, asthma or COPD, and tuberculosis). All statistical analyses were performed using SAS version 9.4 (SAS Institute Inc., Cary, NC, USA).

## 3. Results

### 3.1. Baseline Characteristics

The baseline characteristics of the study population are described in Table 1. During a mean follow-up of 8.3 years, 15,069 premenopausal women were diagnosed with bronchiectasis for an incidence rate of 190.2 cases per 100,000 person/years. Compared with women who did not develop bronchiectasis, those who developed bronchiectasis had a higher mean age (46.9 vs. 45.1 years), a lower BMI (22.9 vs. 23.2 kg/m^2^), a higher proportion of current and ex-smokers (5.2% vs. 5.0%), and more comorbidities: dyslipidaemia (12.7 vs. 11.4%), cardiovascular disease (1.8% vs. 0.9%), asthma or COPD (14.4% vs. 8.3%), and tuberculosis (10.9% vs. 4.1%) (*p* < 0.01 for all variables). Among the reproductive factors, premenopausal women who developed bronchiectasis were more likely than those who did not have menarche at an older age (15.4 vs. 15.1 years), who were nulliparous (14.0% vs. 13.3%), and who had breastfed for 0.5–1 year (35.8% vs. 31.3%) (*p* < 0.01 for all variables). OC use did not differ between those two groups (*p* = 0.18).

During a mean follow-up of 8.1 years, 51,669 postmenopausal women were diagnosed with bronchiectasis for an incidence rate of 468.7 cases per 100,000 person/years. Compared with women who did not develop bronchiectasis, those who developed bronchiectasis were more likely to be older (63.1 vs. 61.6 years), current or ex-smokers (4.1% vs. 3.7%), have a lower BMI (23.8 vs. 24.2 kg/m^2^), and have more comorbidities (cardiovascular disease (7.2% vs. 5.6%), asthma or COPD (23.1% vs. 13.0%), and tuberculosis (12.1% vs. 6.2%). However, women who developed bronchiectasis did not exercise regularly compared with those who did exercise regularly (82.6% vs. 81.6%) (*p* < 0.01 for all variables). With respect to reproductive factors, postmenopausal women who developed bronchiectasis were more likely than those who did not have an older age at menarche (16.6 vs. 16.5 years), who were multiparous (92.3% vs. 91.8%), and who had breastfed for ≥ 1 year (73.1% vs. 70.5%) (*p* < 0.01 for all variables). The rate of participants who received HRT ≥ 5 years was significantly higher among those who developed bronchiectasis than among those who did not (3.3% vs. 2.8%, *p* < 0.01).

### 3.2. Reproductive Factors and the Risk of Bronchiectasis in Premenopausal Women

As shown in Table 2, the multivariable Cox regression analyses showed that an earlier menarche correlated with a lower risk of bronchiectasis (adjusted hazard ratio (aHR) (95% CI) for age at menarche: 0.74 (0.67–0.81) for < 12 years; 0.78 (0.75–0.82) for 12–14 years; 0.89 (0.86–0.93) for 14–16 years) in premenopausal women. Other reproductive factors (parity, duration of breastfeeding, and OC use) were not associated with a risk of bronchiectasis. Similarly, the incidence probability (%) of bronchiectasis differed significantly among the premenopausal women with menarche at age ≥16 years, those at 14–16 years, and those at <14 years (*p* for log-rank test < 0.01).

### 3.3. Reproductive Factors and the Risk of Bronchiectasis in Postmenopausal Women

As shown in Table 3, the univariable Cox regression analyses showed that an earlier menarche, a later menopause, and a longer reproductive period reflecting a longer exposure to female sex hormones decreased the risk of bronchiectasis in a dose-dependent manner. As those three variables were highly correlated, only the reproductive period was included in the multivariable analyses. After adjusting for the potential confounders, a longer reproductive period correlated with a lower risk of bronchiectasis (aHR (95% CI) for the reproductive period: 0.97 (0.95–0.99) for 30–35 years; 0.94 (0.91–0.96) for 35–40 years; 0.90 (0.86–0.94) for ≥ 40 years). Similarly, Figure 2B shows that the cumulative incidence probability (%) of bronchiectasis differed significantly among the postmenopausal women with a reproductive period < 30 years, those with a period of 30–35 years, and those with a period ≥ 35 years (*p* for log-rank test < 0.01).

Regarding other reproductive factors and the risk of bronchiectasis, breastfeeding duration < 1 year (aHR (95% CI) = 0.92 (0.87–0.97) for < 0.5 years and 0.93 (0.88–0.97) for 0.5–1 years) compared with no breastfeeding and OC use < 1 year (adjusted HR (95% CI) = 0.97 (0.94–0.99)) compared with no use were associated with a decreased risk of bronchiectasis. In contrast, the duration of HRT dose-dependently increased the risk of bronchiectasis (aHR (95% CI) for HRT duration: 1.16 (1.12–1.19) for < 2 years, 1.17 (1.12–1.23) for 2–5 years, 1.24 (1.18–1.30) for ≥ 5 years).

## 4. Discussion

In this population-based longitudinal cohort study, we investigated the associations between female reproductive factors and the risk of bronchiectasis. Even after adjusting for potential confounders, a shorter lifetime exposure to endogenous female sex hormones (a later menarche in premenopausal women and a later menarche, earlier menopause, and shorter reproductive period in postmenopausal women) correlated with an increased incidence of bronchiectasis.

As the lifetime duration of endogenous oestrogen exposure varies among women and oestrogen is involved in inflammatory signalling in the airways, it can be postulated that lifetime oestrogen exposure affects the development of lung diseases. Accordingly, previous studies evaluated the effects of female reproductive factors on inflammatory lung diseases such as asthma and COPD [1,8,9,10,11,12,13,14]. However, the association between lifetime oestrogen exposure and the development of bronchiectasis has not been elucidated well. Most previous studies provided indirect evidence such as sex disparity in the prevalence of bronchiectasis [2,4,15,16,17,18,19,20,21,22,23]. Others supported that view by showing the prevalence of bronchiectasis in women affected by their reproductive span [19,23]. However, those previous studies used a cross-sectional methodology and did not consider other reproductive factors [2,4,15,16,17,18,19,20,21,22,23]. Thus, the major strengths of our study were that we used a population-based longitudinal cohort and considered the complexity of various reproductive parameters (parity, breastfeeding, OC use, and HRT) to assess the relationship between lifetime oestrogen exposure and the development of bronchiectasis.

Another important advantage of our study was that we considered comorbidity profiles that might affect the development of bronchiectasis in the analyses of the association of female sex hormones with bronchiectasis. Previous studies have shown that airway diseases such as asthma [36,37], COPD [38], and pulmonary tuberculosis [39] are major aetiologies of bronchiectasis. For example, Crimi et al. showed a link between asthma and bronchiectasis by demonstrating that a type 2 inflammation could provide a causative role in the development of bronchiectasis [36,37]. Additionally, a recent longitudinal study showed that bronchiectasis could develop in subjects with COPD who had bronchial infections; post-tuberculosis bronchiectasis is known to be one of the leading aetiologies of bronchiectasis worldwide [38,39]. Furthermore, as subjects with those comorbidities are more likely to have a low BMI, which can affect female reproductive factors [40,41,42,43], there might be skewed outcomes if these comorbidities were not adjusted. Thus, the comprehensive adjustment of these comorbidities highlights the reliability of our analyses.

The most important finding of our study was that we are, to the best of our knowledge, the first to demonstrate the protective effect of endogenous female sex hormones (probably oestrogen effects because oestrogen exposure is longer than progesterone exposure during the female reproductive cycle) against the development of bronchiectasis. Supporting our results, a few previous studies that evaluated the prevalence of bronchiectasis by sex and age found that the prevalence in women increased abruptly around the beginning of the menopause [19,23]. Previous basic research using a murine model of asthma also suggested that oestrogen might have a protective role against bronchiectasis [44,45,46]. In those studies, oestrogen was shown to have an anti-inflammatory role in the airway by reducing leukocyte recruitment, mucus production, and the secretion of pro-inflammatory cytokines [44,45,46]. Thus, the clinical relevance of our study was on the finding that the evaluation of bronchiectasis might be helpful in women with a short reproductive duration and unexplained respiratory symptoms.

Unexpectedly, we found that HRT was associated with an increased risk of bronchiectasis in postmenopausal women. Although the reasons for that were unclear, we suggest that our current results were most likely a reflection of reverse causality. In other words, women with more severe symptoms due to a more substantial deficiency in endogenous sex hormones (more vulnerable to the development of bronchiectasis) might have had received more HRT than those whose milder symptoms represented a tolerable endogenous sex hormone deficiency, as shown in other studies [47,48].

However, there were other possibilities for the positive relationship between HRT and the risk of bronchiectasis in postmenopausal women. First, a timing hypothesis was suggested that the timing of oestrogen replacement treatment could determine whether it has beneficial or harmful effects on other diseases [47,49]. In line with that view, oestrogen was shown to enhance mucus synthesis in human bronchial epithelial cells as well as dehydrate airway surfaces and reduce cilia beat frequency [50], which could contribute to the progression of bronchiectasis. Synthetic medroxyprogesterone acetate, a major component of HRT, could be second possible explanation. A previous study showed that progesterone inhibited the cilia beat frequency in the airway epithelium, which could inhibit mucociliary clearance [50]. In Korea, most drugs prescribed as HRT in 2010 were progesterogenic (53% used an oestrogen–progesterone combination and 40% used Tibolone, which has a weak estrogenic, progestogenic, and androgenic activity [51]) [52]. Unfortunately, our database lacked accurate data on the type of hormone formulation used and the age at which HRT was initiated. Thus, well-designed future studies are needed to confirm the association between HRT and the risk of bronchiectasis.

Although our study provided the important finding that female sex hormones and the development of bronchiectasis were linked, it had a few limitations. First, the diagnosis of bronchiectasis was made by physicians, which is a major limitation of all claims data-based studies. Thus, there might have been an over- or under-estimation of bronchiectasis. Second, information regarding the reproductive factors was obtained from a health questionnaire so recall bias could not be ruled out. Third, the relationship between the reproductive factors that can lead to relatively short-term changes in life endogenous oestrogen exposure (breastfeeding duration or OC use) and the risk of bronchiectasis in postmenopausal women did not show consistent results; breastfeeding < 1 year was associated with a reduced risk of bronchiectasis whereas breastfeeding ≥ 1 year did not show a significant relationship with the risk of bronchiectasis. OC use did not show any relationship with the risk of bronchiectasis in premenopausal women but OC use < 1 year reduced the risk of bronchiectasis in postmenopausal women. Although these inconsistent results might have been influenced by the relatively short-term exposure of those factors, we could not provide a plausible explanation for this phenomenon. Fourth, because this study was conducted in an exclusively Korean population, generalisability might be limited. Fifth, a vitamin D deficiency is an important factor associated with bronchiectasis [53] but we could not include vitamin D levels in the analyses as these data were not available in our database.

## 5. Conclusions

In conclusion, our results suggest that endogenous female sex hormones might have a protective role against the development of bronchiectasis.

## Figures and Tables

**Figure 1 biomedicines-10-00303-f001:**
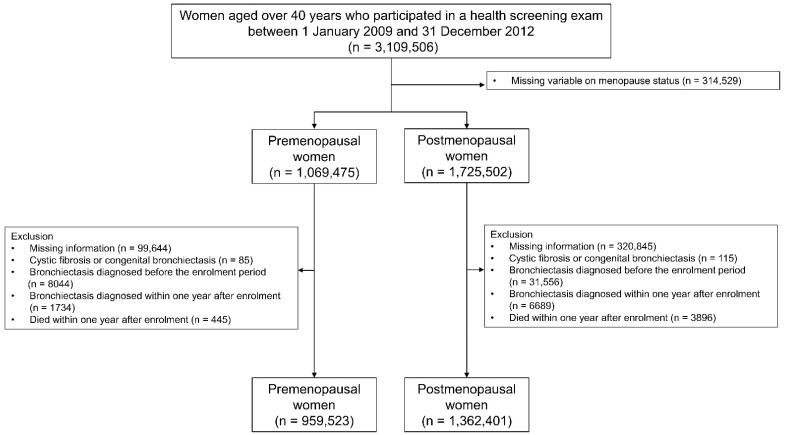
Flow chart of the study population.

**Figure 2 biomedicines-10-00303-f002:**
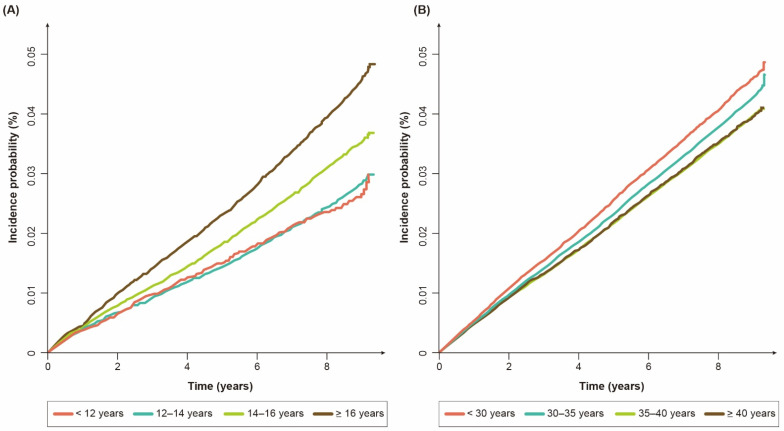
Cumulative incidence probability (%) of bronchiectasis: (**A**) premenopausal women according to age at menarche; (**B**) postmenopausal women according to reproductive period.

**Table 1 biomedicines-10-00303-t001:** Baseline characteristics of the study population.

	Premenopausal Women			Postmenopausal Women		
	Subjects Who Did Not Develop Bronchiectasis (*n* = 944,454)	Subjects Who Did Develop Bronchiectasis (*n* = 15,069)	*p*-Value	Subjects Who Did Not Develop Bronchiectasis (*n* = 1,310,732)	Subjects Who Did Develop Bronchiectasis (*n* = 51,669)	*p*-Value
**Age (years)**	45.1 ± 4.3	46.9 ± 4.9	<0.01	61.6 ± 8.3	63.1 ± 8.2	<0.01
**BMI (kg/m^2^)**	23.2 ± 3.1	22.9 ± 3.1	<0.01	24.2 ± 3.2	23.8 ± 3.3	<0.01
**Smoking history**			<0.01			<0.01
Non-smoker	897,230 (95.0)	14,294 (94.8)		1,262,129 (96.3)	49,536 (95.9)	
Ex-smoker	15,132 (1.6)	213 (1.5)		13,705 (1.1)	600 (1.2)	
Current smoker	32,092 (3.4)	562 (3.7)		34,898 (2.6)	1533 (2.9)	
**Regular physical activity**			0.10			<0.01
No	780,458 (82.6)	12,529 (83.2)		1,069,822 (81.6)	42,643 (82.6)	
Yes	163,996 (17.4)	2540 (16.8)		240,910 (18.4)	9026 (17.4)	
**Income (quartile)**			<0.01			<0.01
Q1 (lowest)	243,033 (25.7)	3913 (25.9)		297,366 (22.7)	11,324 (21.9)	
Q2	191,698 (20.3)	3177 (21.1)		243,680 (18.6)	9305 (18.0)	
Q3	209,064 (22.1)	3405 (22.6)		324,561 (24.8)	12,718 (24.6)	
Q4 (highest)	300,659 (31.9)	4574 (30.4)		445,125 (33.9)	18,322 (35.5)	
**Age at menarche (years)**	15.1 ± 1.7	15.4 ± 1.7	<0.01	16.5 ± 1.8	16.6 ± 1.9	<0.01
**Age at menopause (years)**				50.0 ± 4.1	49.9 ± 4.1	<0.01
**Reproductive period (years)**				33.5 ± 4.4	33.3 ± 4.5	<0.01
**Parity**			<0.01			<0.01
Nulliparous	125,964 (13.3)	2110 (14.0)		30,511 (2.3)	1217 (2.4)	
One child	780,039 (82.0)	12,401 (82.3)		77,231 (5.9)	2738 (5.3)	
Multiparous	38,451 (4.7)	558 (3.7)		1,202,990 (91.8)	47,714 (92.3)	
**Duration of breastfeeding (years)**			<0.01			<0.01
None	230,001 (24.3)	3350 (22.2)		83,450 (6.4)	3175 (6.1)	
<0.5	248,072 (26.2)	3880 (25.8)		81,805 (6.3)	2829 (5.5)	
0.5–1	295,720 (31.3)	5393 (35.8)		220,694 (16.8)	7910 (15.3)	
≥1	170,661 (18.0)	2446 (16.2)		924,783 (70.5)	37,755 (73.1)	
**Duration of OC use (years)**			0.18			<0.01
None	819,633 (86.7)	13,011 (86.3)		1,103,598 (84.2)	43,783 (84.7)	
<1	91,645 (9.7)	1492 (9.9)		123,987 (9.5)	4652 (9.0)	
≥1	33,176 (3.5)	566 (3.8)		83,147 (6.3)	3234 (6.3)	
**Duration of HRT (years)**						<0.01
None				1,104,212 (84.2)	42,837 (82.9)	
<2				120,365 (9.3)	4985 (9.7)	
2–5				49,025 (3.7)	2095 (4.1)	
≥5				37,130 (2.8)	1752 (3.3)	
**Comorbidities**						
Diabetes mellitus	34,637 (3.6)	560 (3.7)	0.75	176,001 (13.4)	6544 (12.6)	<0.01
Dyslipidaemia	108,512 (11.4)	1917 (12.7)	<0.01	452,441 (34.5)	17,336 (33.5)	<0.01
Cardiovascular disease	4669 (0.9)	145 (1.8)	<0.01	50,617 (5.6)	2597 (7.2)	<0.01
Asthma or COPD	78,514 (8.3)	2176 (14.4)	<0.01	171,479 (13.0)	11,949 (23.1)	<0.01
Tuberculosis	38,681 (4.1)	1644 (10.9)	<0.01	82,289 (6.2)	6250 (12.1)	<0.01

Data are presented as n (%) or mean ± standard deviation. BMI: body mass index; OC: oral contraceptive; HRT: hormone replacement therapy; COPD: chronic obstructive pulmonary disease.

**Table 2 biomedicines-10-00303-t002:** HR and 95% CI for the associations between reproductive factors and the risk of bronchiectasis among premenopausal women.

	Total (*n*)	Bronchiectasis (*n*)	Follow-Up Duration (PY)	IR (/1000 PY)	HR (95% CI)	
					Univariable	Multivariable
**Age (years)**					1.07 (1.07, 1.08)	1.07 (1.07, 1.08)
**BMI (kg/m^2^)**					0.97 (0.96, 0.97)	0.95 (0.95, 0.96)
**Smoking history**						
Non-smoker	911,524	14,294	7,528,360.8	1.89	Reference	Reference
Ex-smoker	15,345	213	126,443.2	1.68	0.88 (0.77, 1.01)	0.98 (0.85, 1.12)
Current smoker	32,654	562	268,561.0	2.09	1.10 (1.01, 1.19)	1.14 (1.04, 1.24)
**Regular physical activity**						
No	792,987	12,529	6,545,086.2	1.91	Reference	Reference
Yes	166,536	2540	1,378,278.9	1.84	0.96 (0.92, 1.01)	0.95 (0.91, 1.00)
**Age at menarche (years)**						
<12	42,696	505	352,080.1	1.43	0.56 (0.51, 0.61)	0.74 (0.67, 0.81)
12–14	301,146	3792	2,486,061.3	1.52	0.60 (0.57, 0.62)	0.78 (0.75, 0.82)
14–16	439,342	7059	3,627,513.6	1.94	0.76 (0.73, 0.79)	0.89 (0.86, 0.93)
≥16	176,339	3713	1,457,709.0	2.54	Reference	Reference
**Parity**						
Nulliparous	39,009	558	320,903.4	1.73	Reference	Reference
One child	128,074	2110	1,056,351.2	1.99	1.14 (1.04, 1.26)	1.03 (0.93, 1.14)
Multiparous	792,440	12,401	6,546,110.7	1.89	1.08 (1.00, 1.18)	1.01 (0.92, 1.11)
**Duration of breastfeeding (years)**						
None	173,107	2446	1,427,337.2	1.71	Reference	Reference
<0.5	233,351	3350	1,924,498.6	1.74	1.01 (0.96, 1.07)	0.97 (0.92, 1.03)
0.5–1	251,952	3880	2,081,080.5	1.86	1.08 (1.03, 1.14)	0.99 (0.93, 1.04)
≥1	301,113	5393	2,490,448.7	2.16	1.26 (1.20, 1.32)	1.05 (0.99, 1.11)
**Duration of OC use (years)**						
None	832,644	13,011	6,875,528.4	1.89	Reference	Reference
<1	93,137	1492	769,027.2	1.94	1.02 (0.97, 1.08)	1.04 (0.98, 1.09)
≥1	33,742	566	278,809.5	2.03	1.07 (0.98, 1.16)	1.02 (0.93, 1.11)

Data are presented as n or HR (95% CI). The multivariable model was adjusted for age, BMI, smoking history, regular exercise, income, age at menarche, parity, duration of breastfeeding, duration of OC use, and comorbidities (diabetes mellitus, dyslipidaemia, asthma/COPD, and tuberculosis). PY: person/years; IR: incidence rate; HR: hazard ratio; CI: confidence interval; BMI: body mass index; OC: oral contraceptive; COPD: chronic obstructive pulmonary disease.

**Table 3 biomedicines-10-00303-t003:** HR and 95% CI for the associations between reproductive factors and the risk of bronchiectasis among postmenopausal women.

	Total (*n*)	Bronchiectasis (*n*)	Follow-Up Duration (PY)	IR (/1000 PY)	HR (95% CI)	
					Univariable	Multivariable
**Age (years)**					1.02 (1.02, 1.03)	1.02 (1.02, 1.03)
**BMI (kg/m^2^)**					0.96 (0.95, 0.96)	0.96 (0.96, 0.97)
**Smoking history**						
Non-smoker	1,311,665	49,536	10,623,789.8	4.66	Reference	Reference
Ex-smoker	14,305	600	113,888.1	5.26	1.13 (1.04, 1.22)	1.13 (1.04, 1.23)
Current smoker	36,431	1533	287,390.0	5.33	1.14 (1.08, 1.20)	1.10 (1.05, 1.16)
**Regular physical activity**						
No	1,112,465	42,643	8,985,784.4	4.74	Reference	Reference
Yes	249,936	9026	2,039,283.6	4.42	0.93 (0.91, 0.95)	0.96 (0.94, 0.98)
**Age at menarche (years)**						
<12	12,954	418	105,648.2	3.96	0.79 (0.72–0.87)	
12–14	164,560	5586	1,339,752.9	4.17	0.83 (0.81–0.86)	
14–16	524,836	18,996	4,255,170.5	4.46	0.89 (0.88–0.91)	
≥16	660,051	26,669	5,324,496.4	5.01	Reference	
**Age at menopause (years)**						
<40	24,290	1003	194,175.6	5.17	Reference	
40–45	80,042	3225	641,612.4	5.03	0.97 (0.91, 1.05)	
45–50	371,284	14,078	3,004,544.3	4.69	0.91 (0.85, 0.97)	
50–55	740,042	27,841	5,994,369.0	4.64	0.90 (0.85, 0.96)	
≥55	146,743	5522	1,190,366.4	4.64	0.90 (0.84, 0.96)	
**Reproductive period (years)**						
<30	192,647	8018	1,543,989.8	5.19	Reference	Reference
30–35	568,223	22,028	4,590,126.7	4.80	0.95 (0.90, 0.95)	0.97 (0.95, 0.99)
35–40	513,918	18,479	4,179,829.6	4.42	0.85 (0.83, 0.87)	0.94 (0.91, 0.96)
≥40	87,613	3144	711,121.8	4.42	0.85 (0.82, 0.89)	0.90 (0.86, 0.94)
**Parity**						
Nulliparous	31,728	1217	257,028.2	4.73	Reference	Reference
One child	79,969	2738	650,986.6	4.21	0.89 (0.83, 0.95)	0.95 (0.88, 1.02)
Multiparous	1,250,704	47,714	10,117,053.2	4.72	0.99 (0.94, 1.05)	0.98 (0.92, 1.04)
**Duration of breastfeeding (years)**						
None	86,625	3175	703,992.0	4.51	Reference	Reference
<0.5	84,634	2829	690,832.9	4.10	0.91 (0.86, 0.96)	0.92 (0.87, 0.97)
0.5–1	228,604	7910	1,860,046.3	4.25	0.94 (0.91, 0.98)	0.93 (0.88, 0.97)
≥1	962,538	37,755	7,770,196.7	4.86	1.08 (1.04, 1.12)	0.98 (0.94, 1.02)
**Duration of OC use (years)**						
None	1,147,381	43,783	9,274,646.3	4.72	Reference	Reference
<1	128,639	4652	1,048,247.7	4.44	0.94 (0.91, 0.97)	0.97 (0.94, 0.99)
≥1	86,381	3234	702,174.0	4.61	0.98 (0.94, 1.01)	0.98 (0.94, 1.01)
**Duration of HRT (years)**						
None	1,147,049	42,837	9,265,189.4	4.62	Reference	Reference
<2	125,350	4985	1,025,341.5	4.86	1.05 (1.02, 1.08)	1.16 (1.12, 1.19)
2–5	51,120	2095	417,429.4	5.02	1.09 (1.04, 1.13)	1.17 (1.12, 1.23)
≥5	38,882	1752	317,107.6	5.52	1.20 (1.14, 1.25)	1.24 (1.18, 1.30)

Data are presented as n or HR (95% CI). The multivariable model was adjusted for age, BMI, smoking history, regular exercise, income, reproductive period, parity, duration of breastfeeding, duration of OC use, duration of HRT, and comorbidities (hypertension, diabetes mellitus, dyslipidaemia, asthma/COPD, and tuberculosis). PY: person/years; IR: incidence rate; HR: hazard ratio; CI: confidence interval; BMI: body mass index; OC: oral contraceptive; HRT: hormone replacement therapy; COPD: chronic obstructive pulmonary disease.

## Data Availability

Not applicable.

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
