# Peer review of "Female Reproductive Factors and the Risk of Bronchiectasis: A Nationwide Population-Based Longitudinal Study"

_biomedicines, 2022, doi:10.3390/biomedicines10020303_

Round 1

Reviewer 1 Report

The most important strength of this study is the number of patients analyzed. The single-payer universal health system of Korea is fantastic for collecting data at national level and can shed light on the incidence
and patient characteristics. Furthermore, these are patients who come from all socio-economic categories, and this improves the external validity of the results.
An additional interesting factor of this study is that very few studies have evaluated the protective effect of endogenous female sex hormones against development of bronchiectasis. Some previous studies found a sex disparity in the prevalence of bronchiectasis, and suggest a close relationship between female sex hormones and the risk to develope bronchiectasis, but no study has considered all the female reproductive factors, like in this study.
Furthermore, analysis of the associations between female reproductive factors and risk of bronchiectasis in premenpausal women and postmenopausal women at international level and comparison with results from other countries can improve our knowledge of this association.

I believe a major problem of this study is that diagnosis of bronchiectasis was made by physicians, using ICD-10 codes, and this is a major limitation of all claims data–based studies. In this way there are doubts about the accuracy of the diagnosis of bronchiectasis.

Author Response

## Response to Reviewer 1’s comments

General comments

The most important strength of this study is the number of patients analyzed. The single-payer universal health system of Korea is fantastic for collecting data at national level and can shed light on the incidence and patient characteristics. Furthermore, these are patients who come from all socio-economic categories, and this improves the external validity of the results. An additional interesting factor of this study is that very few studies have evaluated the protective effect of endogenous female sex hormones against development of bronchiectasis. Some previous studies found a sex disparity in the prevalence of bronchiectasis, and suggest a close relationship between female sex hormones and the risk to develope bronchiectasis, but no study has considered all the female reproductive factors, like in this study.

Furthermore, analysis of the associations between female reproductive factors and risk of bronchiectasis in premenpausal women and postmenopausal women at international level and comparison with results from other countries can improve our knowledge of this association.

Response. Thank you for your positive comments. Regarding the concerns raised by the reviewer, we provided point-by-point responses as below.

I believe a major problem of this study is that diagnosis of bronchiectasis was made by physicians, using ICD-10 codes, and this is a major limitation of all claims data–based studies. In this way there are doubts about the accuracy of the diagnosis of bronchiectasis.
Response. We fully agree with your opinion. The diagnosis of bronchiectasis was made by physicians, which is a major limitation of all claims data-based studies. Thus, there might have been over or under-estimation of bronchiectasis. We added this as a limitation of our study.

[Revised] p. 9, lines 292-294 in the revised manuscript

First, diagnosis of bronchiectasis was made by physicians, which is a major limitation of all claims data-based studies. Thus, there might have been over or under-estimation of bronchiectasis.

Reviewer 2 Report

I would like to thank the authors for having shown interest in this topic with this article which focuses on the importance of reproductive factors as an entity of risk for the development of bronchiectasis.
The work is relevant but nevertheless some points emerge that in my opinion should be highlighted more.
1) It is now known in the literature and in clinical practice how important the association between bronchiectasis (BE) and bronchial asthma is. In the methods section it would be necessary to deepen this association between comorbidities and perhaps expose in the discussion section what emerges from the Odeirna literature and add the following references: a) The Link between Asthma and Bronchiectasis: State of the Art. Respiration. 2020; 99 (6): 463-476. doi: 10.1159 / 000507228. Epub 2020 May 28. PMID: 32464625 and b) Type 2-High Severe Asthma with and without Bronchiectasis: A Prospective Observational Multicentre Study. J Asthma Allergy. 2021 Nov 30; 14: 1441-1452. doi: 10.2147 / JAA.S332245. PMID: 34880630; PMCID: PMC8646229.

2) Another important missing evaluation is the link between bronchiectasis and vitamin D deficiency which, especially in the post-menopausal age, is very evident. It would be useful in the Methods section to enter, if present in the collection of patient characteristics, the vitamin D level of these patients. In this regard it would be necessary to include in the references a) Vitamin D and disease severity in bronchiectasis. Respir Med. 2019 Mar; 148: 1-5. doi: 10.1016 / j.rmed.2019.01.009. Epub 2019 Jan 24. PMID: 30827468.

Author Response

## Response to Reviewer 2’s comments

General comments

I would like to thank the authors for having shown interest in this topic with this article which focuses on the importance of reproductive factors as an entity of risk for the development of bronchiectasis. The work is relevant but nevertheless some points emerge that in my opinion should be highlighted more.

Response. Thank you for your positive comments. Regarding the concerns raised by the reviewer, we provided point-by-point responses as below.

Specific comments

C1. It is now known in the literature and in clinical practice how important the association between bronchiectasis (BE) and bronchial asthma is. In the methods section it would be necessary to deepen this association between comorbidities and perhaps expose in the discussion section what emerges from the Odeirna literature and add the following references: a) The Link between Asthma and Bronchiectasis: State of the Art. Respiration. 2020; 99 (6): 463-476. doi: 10.1159 / 000507228. Epub 2020 May 28. PMID: 32464625 and b) Type 2-High Severe Asthma with and without Bronchiectasis: A Prospective Observational Multicentre Study. J Asthma Allergy. 2021 Nov 30; 14: 1441-1452. doi: 10.2147 / JAA.S332245. PMID: 34880630; PMCID: PMC8646229.

R1. We thank the reviewer for this valuable comment that asthma is associated with bronchiectasis. According to the reviewer's comments, we now have deepened the association between pulmonary comorbidities including asthma and bronchiectasis in the Method section. We provided information that pulmonary comorbidities including asthma were adjusted since these comorbidities affect the development of bronchiectasis. Also, we discussed this association in the discussion section and added the references that the reviewer suggested.

[Revised] p. 4, lines 133136 in the revised manuscript

Baseline demographics, metabolic comorbidities that might interact with female reproductive factors (diabetes mellitus and dyslipidaemia), and pulmonary comorbidities that might affect the development of bronchiectasis (e.g., asthma [36,37], COPD [38], and tuberculosis [39]) were considered in the multivariable analyses.

[Revised] p. 9, lines 243-255 in the revised manuscript

Another important advantage of our study is that we considered comorbidities profiles that might affect the development of bronchiectasis into the analyses of the association of female sex hormones with bronchiectasis. Previous studies have shown that airway diseases, such as asthma [36,37] and COPD [38], and pulmonary tuberculosis [39] are major aetiologies of bronchiectasis. For example, Crimi et al. showed the link between asthma and bronchiectasis by showing type 2 inflammation could provide a causative role in the development of bronchiectasis [36,37]. Additionally, a recent longitudinal study has shown that bronchiectasis can be developed in subjects with COPD who had bronchial infections, and post-tuberculosis bronchiectasis is known to be one of the leading aetiologies of bronchi-ectasis worldwide [38,39]. Furthermore, since subjects with those comorbidities are more likely to have low BMI which can affect female reproductive factors [40-43], there might be skewed outcomes if these comorbidities were not adjusted. Thus, the comprehensive adjustment of those comorbidities highlights the reliability of our analyses.

[References for revision]

  1. Crimi, C.; Ferri, S.; Campisi, R.; Crimi, N. The Link between Asthma and Bronchiectasis: State of the Art. Respiration; international review of thoracic diseases 2020, 99, 463-476, doi:10.1159/000507228.
  2. Crimi, C.; Campisi, R.; Nolasco, S.; Ferri, S.; Cacopardo, G.; Impellizzeri, P.; Pistorio, M.P.; Fagone, E.; Pelaia, C.; Heffler, E.; et al. Type 2-High Severe Asthma with and without Bronchiectasis: A Prospective Observational Multicentre Study. Journal of asthma and allergy 2021, 14, 1441-1452, doi:10.2147/jaa.S332245.
  3. Martínez-García, M.; de la Rosa-Carrillo, D.; Soler-Cataluña, J.J.; Catalan-Serra, P.; Ballester, M.; Roca Vanaclocha, Y.; Agramunt, M.; Ballestin, J.; Garcia-Ortega, A.; Oscullo, G.; et al. Bronchial Infection and Temporal Evolution of Bronchiectasis in Patients With Chronic Obstructive Pulmonary Disease. Clin Infect Dis 2021, 72, 403-410, doi:10.1093/cid/ciaa069.
  4. Chandrasekaran, R.; Mac Aogáin, M.; Chalmers, J.D.; Elborn, S.J.; Chotirmall, S.H. Geographic variation in the aetiology, epidemiology and microbiology of bronchiectasis. BMC Pulm Med 2018, 18, 83, doi:10.1186/s12890-018-0638-0.
  5. Zhu, D.; Chung, H.F.; Pandeya, N.; Dobson, A.J.; Kuh, D.; Crawford, S.L.; Gold, E.B.; Avis, N.E.; Giles, G.G.; Bruinsma, F.; et al. Body mass index and age at natural menopause: an international pooled analysis of 11 prospective studies. Eur J Epidemiol 2018, 33, 699-710, doi:10.1007/s10654-018-0367-y.
  6. Casha, A.R.; Scarci, M. The link between tuberculosis and body mass index. Journal of thoracic disease 2017, 9, E301-e303, doi:10.21037/jtd.2017.03.47.
  7. McDonald, M.N.; Wouters, E.F.M.; Rutten, E.; Casaburi, R.; Rennard, S.I.; Lomas, D.A.; Bamman, M.; Celli, B.; Agusti, A.; Tal-Singer, R.; et al. It's more than low BMI: prevalence of cachexia and associated mortality in COPD. Respir Res 2019, 20, 100, doi:10.1186/s12931-019-1073-3.
  8. Sio, Y.Y.; Chew, F.T. Risk factors of asthma in the Asian population: a systematic review and meta-analysis. Journal of physiological anthropology 2021, 40, 22, doi:10.1186/s40101-021-00273-x.

C2. Another important missing evaluation is the link between bronchiectasis and vitamin D deficiency which, especially in the post-menopausal age, is very evident. It would be useful in the Methods section to enter, if present in the collection of patient characteristics, the vitamin D level of these patients. In this regard it would be necessary to include in the references a) Vitamin D and disease severity in bronchiectasis. Respir Med. 2019 Mar; 148: 1-5. doi: 10.1016 / j.rmed.2019.01.009. Epub 2019 Jan 24. PMID: 30827468.
R2. Thank you for your helpful comments. As the reviewer commented, it would be more informative if we could have included vitamin D in our analyses. However, unfortunately, our database does not have vitamin D levels. Thus, we could not evaluate this association. We added this as a limitation in the revised manuscript.

[Revised] p. 10, lines 306–308 in the revised manuscript

Fifth, vitamin D deficiency is an important factor associated with bronchiectasis [53]. However, we could not include vitamin D levels in the analyses since this data was not available in our database.

[Reference for revision]

  1. Ferri, S.; Crimi, C.; Heffler, E.; Campisi, R.; Noto, A.; Crimi, N. Vitamin D and disease severity in bronchiectasis. Respir Med 2019, 148, 1-5.

Reviewer 3 Report

Major Issues:
- I believe a major limitation of this study, as also reported in the discussion of the article, is that diagnosis of bronchiectasis was made by physicians, using ICD-10 codes, and this is a major limitation of all claims data–based studies. In this way there are doubts about the accuracy of the diagnosis of bronchiectasis.
- Another limitation is about the information of the reproductive factors that was obtain from a health questionnaire, so memory errors or errors in marking the right answer cannot be ruled out.

Minor Issues:
- Line 100-101: “The duration of HRT was categorized as none, < 2 years, 2-4 years, and ≥ 5 years”, shouldn’t the third groud be “> 4 years” instead of “≥ 5 years”? It’s not clear the subdivision of these ranges.

-Line 45: “Female sex hormones have been suggested to contribute to this sex disparity, and many previous studies have evaluated the associations between female sex hormones and development of chronic airway
diseases”. In this regard, please cite a study of Scioscia et al. (J Clin
Med. 2020 Jun 29;9(7):2037. doi: 10.3390/jcm9072037) that has seen how, in a common chronic airway disease like asthma, the severity differs according to gender, and in adult women, there is higher prevalence and severity of asthma than in men, and it coincides with changes in sex hormones. The 17β-estradiol (E2) concentrations in the blood and airways of women affected by asthma onset after menopause, evaluating its possible role in the severity of the disease.

Author Response

## Response to Reviewer 3’s comments

Specific comments

C1. I believe a major limitation of this study, as also reported in the discussion of the article, is that diagnosis of bronchiectasis was made by physicians, using ICD-10 codes, and this is a major limitation of all claims data–based studies. In this way there are doubts about the accuracy of the diagnosis of bronchiectasis.

R1. We fully agree with your opinion. The diagnosis of bronchiectasis was made by physicians, which is a major limitation of all claims data-based studies. Thus, there might have been over or under-estimation of bronchiectasis. We added this as a limitation of our study in the revised manuscript.

[Revised] p. 10, lines 306–308 in the revised manuscript

First, diagnosis of bronchiectasis was made by physicians, which is a major limitation of all claims data-based studies. Thus, there might have been over or under-estimation of bronchiectasis.

C2. Another limitation is about the information of the reproductive factors that was obtain from a health questionnaire, so memory errors or errors in marking the right answer cannot be ruled out.

R2. We agree with the limitation you suggested. We mentioned this in the second part of the limitations section.

C3. Line 100-101: “The duration of HRT was categorized as none, < 2 years, 2-4 years, and ≥ 5 years”, shouldn’t the third groud be “> 4 years” instead of “≥ 5 years”? It’s not clear the subdivision of these ranges.

R3. Thank you for pointing out the error. The duration of HRT was categorized as none, < 2 years, 2-5 years (≥ 2 and < 5 years), and ≥ 5 years. We corrected “2–4 years” to “2–5 years” in the revised manuscript (please see lines 109-111). We also clarified the categories of other reproductive factors (please see lines 99–111).

C4. Line 45: “Female sex hormones have been suggested to contribute to this sex disparity, and many previous studies have evaluated the associations between female sex hormones and development of chronic airway diseases”. In this regard, please cite a study of Scioscia et al. (J Clin Med. 2020 Jun 29;9(7):2037. doi: 10.3390/jcm9072037) that has seen how, in a common chronic airway disease like asthma, the severity differs according to gender, and in adult women, there is higher prevalence and severity of asthma than in men, and it coincides with changes in sex hormones. The 17β-estradiol (E2) concentrations in the blood and airways of women affected by asthma onset after menopause, evaluating its possible role in the severity of the disease.

R4. We thank the reviewer for this valuable comment. We have cited the study in the revised manuscript.

[Revised] p. 2, lines 47–50 in the revised manuscript

Female sex hormones including 17ß-estradiol have been suggested to contribute to this sex disparity, and many previous studies have evaluated the associations between female sex hormones and development of chronic airway diseases [1,2,7-14].

[Reference for revision]

  1. Scioscia, G., et al., The Role of Airways 17beta-Estradiol as a Biomarker of Severity in Postmenopausal Asthma: A Pilot Study. J Clin Med, 2020. 9(7).